# High-Temperature Creep and Microstructure Evolution of Alloy 800H Weldments with Inconel 625 and Haynes 230 Filler Materials

**Wenjing Li *** , **Lin Xiao, Lori Walters, Qingshan Dong** , **Maurizio Ienzi and Robyn Sloan**

Canadian Nuclear Laboratories, Chalk River, ON K0J 1J0, Canada; lin.xiao@cnl.ca (L.X.);
lori.walters@cnl.ca (L.W.); qingshan.dong@cnl.ca (Q.D.); maurizio.ienzi@cnl.ca (M.I.); robyn.sloan@cnl.ca (R.S.)
* Correspondence: wenjing.li@cnl.ca

**Abstract:** Alloy 800H stands as one of the few code-qualified materials for fabricating in-core and out-of-core components operating in high-temperature reactors. Welding is a common practice for assembling these components; however, the selection of a suitable filler material is essential for enhancing the high-temperature creep resistance of Alloy 800H weldments in high-temperature applications. In this study, Inconel 625 and Haynes 230 filler materials were used to weld Alloy 800H plates by employing the gas tungsten arc welding technique. The high-temperature tensile and creep rupture properties, microstructural stability, and evolution of the weldments after high-temperature exposure were investigated and compared with those of Alloy 800H. The results show that both weldments exhibit enhanced tensile and creep behavior at 760 °C. The creep rupture times of the weldments with Inconel 625 filler and Haynes 230 filler materials were about two and three time longer, respectively, than those of Alloy 800H base metal when tested at 80 MPa and 760 °C. Carbides (MC and $M_{23}C_6$) were commonly observed in the microstructures of both the weld and base metals in the two weldments after high-temperature creep tests. However, the Inconel 625 filler weldment displayed detrimental δ and Laves phases in the fusion zone, and these precipitates could be potential sites for initiating cracks following prolonged high-temperature exposure. This study shows that the weldment with Haynes 230 filler material exhibit better phase stability and creep rupture properties than the one with Inconel 625, suggesting its potential for use as a candidate filler material for Alloy 800H for further investigation. This finding also emphasizes the critical consideration of microstructural evolutions and phase stability in evaluating high-temperature materials and their weldments in high-temperature reactor applications.

**Keywords:** Alloy 800H; Inconel 625; Haynes 230; creep; microstructure; weldment

## 1. Introduction

Incoloy 800H (or Alloy 800H, Unified Numbering System No. N08810) is one of a few code-qualified materials used at temperatures up to 760 °C for fabricating in-core and out-of-core components operating in high-temperature reactors, e.g., reactivity control rods and intermediate heat exchangers. The alloy is an iron-based alloy with a nominal composition of Fe-32.5Ni-21Cr [1]. To improve the creep rupture properties, Alloy 800H maintains a carbon content of at least 0.05%. Alloy 800H must undergo solution annealing at temperatures above 1050 °C, such as 1093 °C, to achieve a stable austenitic structure. It is intended for use at temperatures > 593 °C [1]. The annealing treatment at temperatures above 1050 °C is known to produce an ASME grain size number of 5 or coarser, i.e., coarser than 65 μm. This alloy is currently approved under ASME Section III, Division 5 for nuclear service up to 760 °C [2]. There are efforts to extend the ASME-allowable stresses of Alloy 800H beyond 800 °C, preferably up to 950 °C, and to achieve a service lifetime of 60 years. These efforts are based on the concepts of longer lifetimes and higher operation

temperatures, including under off-normal and accident conditions, for various components used in the development of modern high-temperature gas-cooled reactors [3].

Welding processes are required for assembling components in the application of Alloy 800H. Achieving high reliability in a long-term, high-temperature service environment requires the welds to possess high creep resistance [4]. However, a suitable filler material to improve the high-temperature creep resistance of the Alloy 800H weldments is required in the high-temperature application. Cross-weld creep testing of most Ni-based Inconel and stainless steel alloys indicates that weld strengths are usually in the range of 50–100% of those of the Alloy 800H base material [4–6]. Commonly used filler materials for Alloy 800H welding, such as Inconel 82, 182, 188, and SS 304 alloys, exhibit slightly lower creep rupture stresses than Alloy 800H base metal, in the range of 538 to 760 °C, regardless of whether the exposure time is short or long [5,7,8]. A recent investigation at the Idaho National Laboratory using Inconel 617 filler material did not demonstrate an improvement in creep rupture properties [9]. In addition, the high content of cobalt in Alloy 617 (10–15 wt.%) makes it unsuitable due to the activation of the cobalt under neutron irradiation. Haynes 230, a solid-solution-strengthened Ni-Cr-W alloy, exhibits more enhanced creep rupture strength than Alloy 800H at temperatures higher than 475 °C. It also demonstrates better corrosion resistance at elevated temperatures and contains a lower cobalt concentration [10]. Inconel 625, a Ni-Cr-Mo alloy with additional Nb, has an excellent combination of corrosion resistance and superior creep resistance at high temperatures over 600 °C [11,12]. In this paper, Inconel 625 and Haynes 230 were selected as candidate filler materials to produce Alloy 800H weldments (WMs) using the gas tungsten arc welding technique. The high-temperature tensile and creep properties of these weldments were then compared and analyzed. The microstructures before and after the creep test and fracture mode were reported. The creep mechanisms, dislocation structure, and precipitates evolution were explicated.

## 2. Materials and Experiments

### 2.1. Materials

Alloy 800H plates with dimensions of 508 mm (20 in) length, 152 mm (6 in) width, and 19 mm (3/4 in) thickness were used in this study. The plates were melted and manufactured by VDM Metals, Werdohl, Germany. The material was solution-annealed at 1167 °C for 32 min according to the specification provided by the supplier. The as-received plates had equiaxial austenitic structures with an average grain size of 121 ± 78 μm, which is within the specification of Alloy 800H (ASME grain size number of 5 or coarser [4]). Inconel 625 and Haynes 230 filler wires each had a diameter of 1.6 mm (0.0625 in). The compositions of the as-received 800H and the nominal chemical compositions of the two filler materials are provided in Table 1.

**Table 1.** Chemical composition of as-received (AR) Alloy 800H and nominal chemical compositions of Inconel 625 and Haynes 230 filler materials (wt.%) [13,14].

| Element | Fe | Ni | Cr | Mo | Nb | Co | Mn | C | Al | Ti | Si | B | W |
|---|---|---|---|---|---|---|---|---|---|---|---|---|---|
| AR Alloy 800H | 45.6 | 30.3 | 20.6 | 0.7 | - | 0.05 | 0.7 | 0.08 | 0.49 | 0.52 | 0.4 | - | - |
| Inconel 625 | <5.0 | >58.0 | 20.0 -23.0 | 8.0 -10.0 | 3.15 -4.15 | <1.00 | <0.50 | <0.10 | <0.40 | <0.40 | <0.50 | - | - |
| Haynes 230 | <3.0 | 57.0 Bal | 22.0 | 2.0 | <0.50 | <5.00 | 0.50 | 0.10 | 0.30 | <0.10 | 0.40 | <0.015 | 14.0 |

### 2.2. Welding Process and Specimen Fabrication

The gas tungsten arc welding technique was employed in the welding process. A double-sided modified X-type groove profile butt joint was designed for fabricating the Alloy 800H weldments (Figure 1a). Multiple passes were performed perpendicularly to the rolling direction (Figure 1b). Minimum distortion after the welding process was achieved

and no post-weld heat treatment was performed. Bend testing was performed, and all of the bent samples failed outside of the fusion zones.

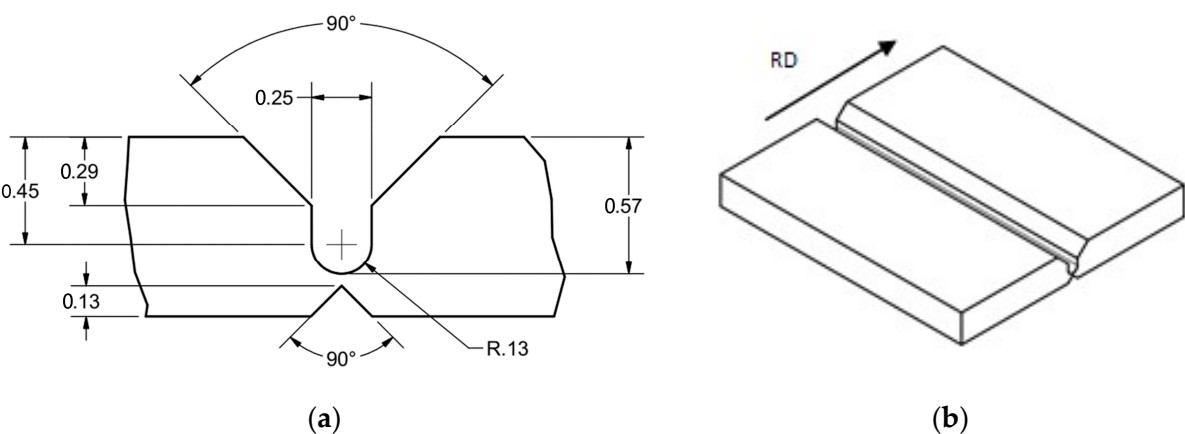

(**a**)                                                                                            (**b**)

**Figure 1.** Groove profile of the butt joint. (**a**) Schematic view of cross section and geometry, (**b**) schematic view of the plate with a groove showing the rolling direction (RD) (all dimensions are in inches).

Rod-shaped tensile and creep specimens with a diameter of 0.25 in (6.35 mm) were machined across the weld beam, and the pulling direction was aligned with the rolling direction of the plates. Figure 2 provides the geometry of the specimens. Note that the U-shaped section of the weld seam (U-shaped notch, as shown in Figure 1a) was positioned in the middle of the gauge section of the specimens. The fusion zone width was, therefore, relatively uniform, with a width of approximately 0.25 in (6.35 mm) as indicated by the shaded area in Figure 2. The 60° V-shaped and 0.030 in wide grooves beside the shoulders were used to insert the extensometer crossheads for creep strain measurements.

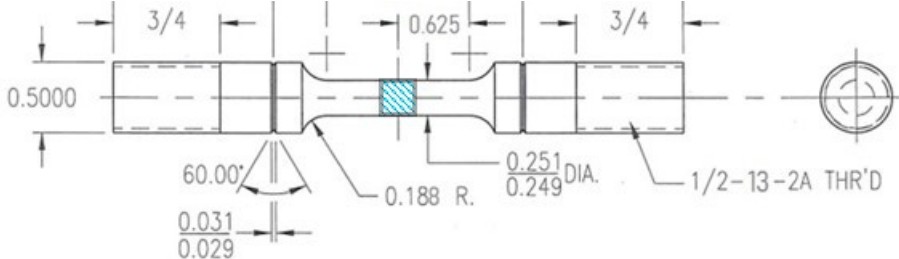

**Figure 2.** Geometry of tensile and creep specimens (all dimensions are in inches). The location of the fusion zone (blue shaded area, about 0.25 in or 6.35 mm) is in the center of the reduced area section.

### 2.3. Hardness Measurement

The Vickers hardness of the welds was measured using a Struers hardness tester, Struers, Cleveland, OH, USA. A load of 500 g force (gf) with a dwell time of 15 s was used. After mechanical grinding using a series of silicon sandpapers, the specimens were finely polished with 0.5 μm diamond suspensions and colloidal silica. The polished samples were then etched with Marble's reagent (10 mL $H_2O$ + 50 mL HCl + 10 mL $CuSO_4$). The measurement was performed across the weld metal, fusion boundary, heat-affected zone, and unaffected base metal at distance intervals of 300 μm, and the average value of three measurements at each location was reported.

### 2.4. High-Temperature Tensile and Creep Tests

High-temperature tensile tests were conducted at 760 °C in a hydraulic MTS testing system (Model 318.25), MTS, Eden-Prairie, MN, USA, with a force capacity of 250 kN. A three-zone MTS high-temperature split furnace (Model 653.04B), MTS, Eden-Prairie, MN, USA, mounted on the tester was used to heat the specimens and maintain a constant

temperature. Temperature was monitored with three K-type thermocouples in contact with the top, middle, and bottom of the specimen's reduced area section, and the temperature profile remained constant within $\pm 1\,°C$ of the nominal temperature during the test. A strain rate of $1 \times 10^{-4}\,s^{-1}$ was used, and the soak time was set to 30 min at the test temperature before testing. A high-temperature self-supporting furnace extensometer (Epsilon Model 3448), Epsilon, Jackson, WY, USA, was used to measure displacement.

High-temperature creep tests were performed at $760\,°C$ in air using creep testers from Applied Test Systems (ATS, Butler, PA, USA). Load was applied using calibrated weights through a counterbalanced lever arm with a ratio of 3:1. A three-zone ATS Series 3210 split tube furnace was mounted on the creep tester to heat the specimens and maintain a constant temperature. Temperature was monitored with three K-type thermocouples in contact with the top, middle, and bottom of the specimen's reduced area section. The test temperature was controlled within $\pm 0.5\,°C$ of the target temperature without overshooting. The WinSCC software was used to control and monitor the tests, and to collect and export experimental data. The creep specimen was heated to the target temperature and soaked for 1 h before the load was applied for testing. Strain was measured with the HEIDENHAIN dual averaging linear displacement encoder connected to a Series 4124 average extensometer frame. The measurement accuracy was $\pm 0.2\,\mu m$. The adjusted length of the reduced area section was used to calculate the creep strain according to ASTM E139 [15]. The specimen was tested until rupture.

### 2.5. Microstructure Examination

As-welded and post-creep test microstructures were characterized. Transmission electron microscopy (TEM) analyses were performed at the different locations of the crept specimen, namely, the fusion zone (FZ) weld metal, heat-affected zone (HAZ) adjacent to the fusion boundary, and the base metal (BM) close to the fracture surface in the gauge section, to analyze the plastic deformation behavior and phase evolutions in each region. The microstructure of the threaded end section was also analyzed. This section experienced negligible deformation but had the same thermal exposure history as the gauge section, and their comparison would help distinguish the creep deformation mechanism from the high-temperature ageing effect.

The cut slices from the specimens were first mechanically ground to a thickness of 0.2 mm, then electropolishing was conducted at 17 V DC and $-30\,°C$ with an electrolyte composed of 95% methanol and 5% perchloric acid. The electron-transparent foils were examined using FEI OSIRIS scanning TEM (STEM), FEI, Hillsboro, OR, US, operated at 200 kV. A nano-beam electron diffraction (NBED) technique was used to analyze the crystal structure of precipitates. Local chemical compositions were investigated by ChemiSTEM energy-dispersive X-ray spectroscopy (EDS) using the Cliff–Lorimer quantification approach implemented within the Brucker ESPRIT software, Version 1.9. The fracture surface was examined with Hitachi scanning electron microscopy (SEM), Hitachi, Marunouchi, Chiyoda-ku, Tokyo, Japan.

## 3. Results

### 3.1. Hardness

Figure 3 provides plots of the measured Vickers hardness (HV) across the FZ, HAZ, and unaffected base metal of the weldments with the two filler materials. The hardness in the FZ ranged from 170 to 190 HV, which was significantly higher than the approximately 120 HV of the base metal. From the fusion boundary, the hardness experienced a gradual decrease over a span of approximately 7 mm before reaching the hardness of the unaffected BM. In the machined weldment specimens, as illustrated in Figure 2, a 25 mm gauge section consisted of approximately 6.4 mm of weld metal in the middle, 7 mm of HAZ, and 1 to 2 mm of unaffected BM materials on each side of the FZ, i.e., approximately 26% of volume fraction in the FZ, 60% in the HAZ, and 14% in the unaffected BM materials, respectively.

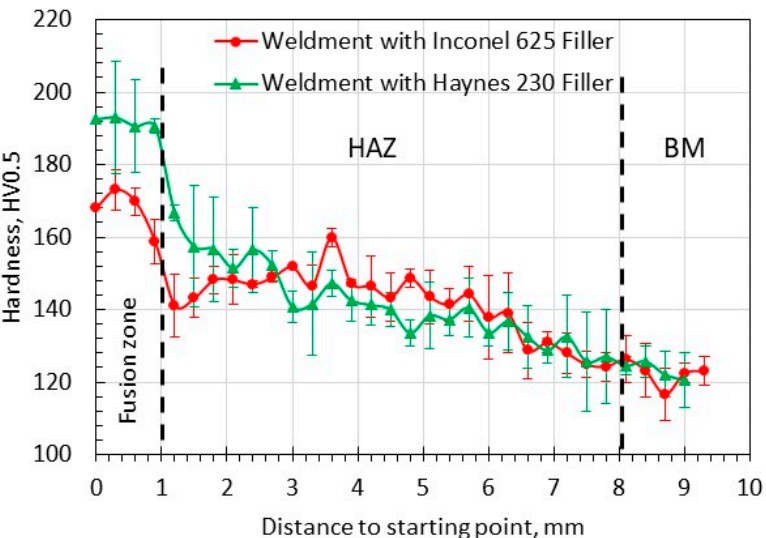

**Figure 3.** Vickers hardness across the FZ, HAZ, and unaffected BM of the weldments. The two dashed vertical lines indicate the approximate boundaries.

### 3.2. High-Temperature Tensile Test

Figure 4 presents the tensile stress–strain curves of the Alloy 800H BM specimen and the weldment specimens with Inconel 625 and Haynes 230 filler materials. The tests were conducted at a strain rate of $1 \times 10^{-4}$ s$^{-1}$ at 760 °C. Both weldment specimens experienced rupture in the HAZ section of the gauge length during the tensile tests. It is noted that the stress–strain curves of the weldments are considered nominal due to material and property heterogeneity in the gauge section. The results show that the apparent 0.2% yield stress (YS) of the weldment specimens is higher than that of Alloy 800H, but the apparent ductility is lower. The apparent YS of welds reached 203 MPa and 194 MPa with the Inconel 625 and Haynes 230 filler materials, compared to 118 MPa with the BM specimen. The higher apparent YS of the weldment specimens was likely due to the higher hardness of the FZ and HAZ materials, which constituted a total volume fraction of approximately 86% in the gauge section. The ultimate tensile strengths (UTSs) of the welds are similar to that of BM but slightly higher, i.e., 236 MPa and 238 MPa for the WMs with Inconel 625 and Haynes 230 filler materials, respectively, versus 224 MPa for the 800H. The comparable UTS between the BM and the weldment specimens and the fracture locations in the HAZ indicated a satisfactory weld quality in this study.

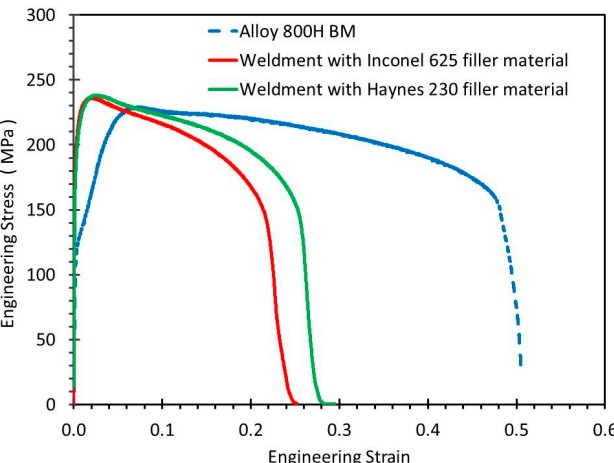

**Figure 4.** Tensile stress–strain curves of Alloy 800H BM and the weldment specimens with Inconel 625 and Haynes 230 filler materials at a strain rate of $1 \times 10^{-4}$ s$^{-1}$ at 760 °C.

### 3.3. High-Temperature Creep Test

Figure 5a,b provides the creep strains and strain rates at a constant applied load of 80 MPa at 760 °C as a function of creep time in the Alloy 800H BM and the weldment specimens. The onset of tertiary creep was defined by the 0.2% strain offset from linear secondary creep (Figure 5b), following the criteria in [16]. Table 2 lists the minimum creep strain rate, creep strain at rupture, time to tertiary stage ($T_t$), and creep time at rupture ($T_R$). The apparent minimum creep strain rates of the weldment specimens were about one order of magnitude lower than that of the BM specimen. The time to tertiary stage and the time to rupture of the BM specimen were much shorter than those of the weldments. Furthermore, the apparent creep strains of the weldment specimens at rupture were only about 10%; in contrast, the creep strain of the 800H BM specimen reached as high as 45%. In the weldment specimens, the ratios between the time to tertiary stage and time to rupture were close to 0.3, and the ratio for the Alloy 800H BM specimen was only 0.13, indicating that the majority of the creep time was in the tertiary stage in both the BM and the weldment specimens.

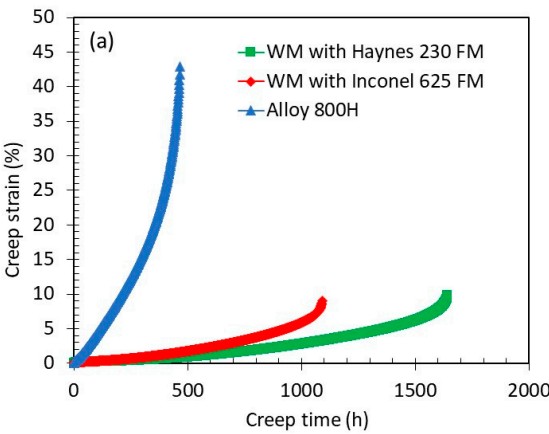 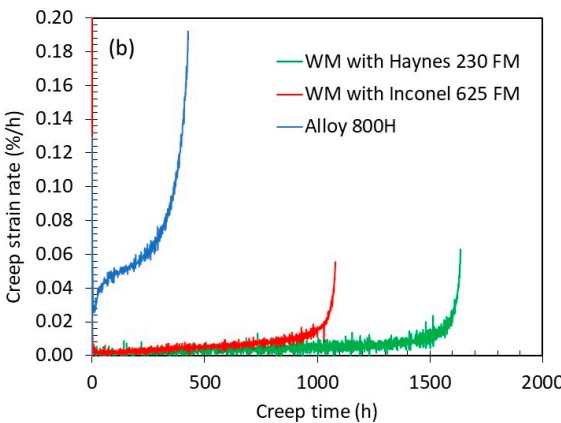

**Figure 5.** Creep curves of the Alloy 800H BM and WM specimens with Inconel 625 and Haynes 230 filler materials at 760 °C and 80 MPa. (**a**) Creep strain. (**b**) Creep strain rate as a function of time.

**Table 2.** Creep strain parameters of Alloy 800H and the weldments at 760 °C and 80 MPa.

| Material | Minimum Strain Rate ($h^{-1}$) | Strain at Rupture (%) | Time to Tertiary Stage (h), $T_t$ | Time to Rupture (h), $T_R$ |
|---|---|---|---|---|
| Incoloy 800H BM | $3.51 \times 10^{-4}$ | 44.96 | 60 | 467 |
| Weldment with Inconel 625 filler | $1.89 \times 10^{-5}$ | 9.66 | 320 | 1091 |
| Weldment with Haynes 230 filler | $1.04 \times 10^{-5}$ | 10.51 | 456 | 1643 |

Figure 6 provides images of the ruptured BM and weldment specimens after the creep tests. The base metal specimen was relatively uniform within the gauge section, with slight necking observed near the fracture surface (Figure 6a). Surface cracks were evenly distributed along the gauge section and were perpendicular to the applied loading direction. In both weldment specimens, the FZ and the HAZ adjacent to the fusion boundary experienced relatively uniform deformation, with no surface cracks in these areas. Necking and rupture occurred closed to the unaffected base metal, as shown in Figure 6b,c. This localized deformation was attributed to the lower hardness of the unaffected base metal compared to the FZ and HAZ, as illustrated in Figure 3. The materials were cut from different locations in the tested weldment specimen, labeled as 1, FZ; 2, HAZ adjacent to the fusion boundary; 3, base metal region close to rupture surface; and 4, un-deformed region in the threaded section. Subsequent sections in this paper will present and compare the microstructures of as-received 800H BM and as-welded materials, as well as the post-creep weldment specimens.

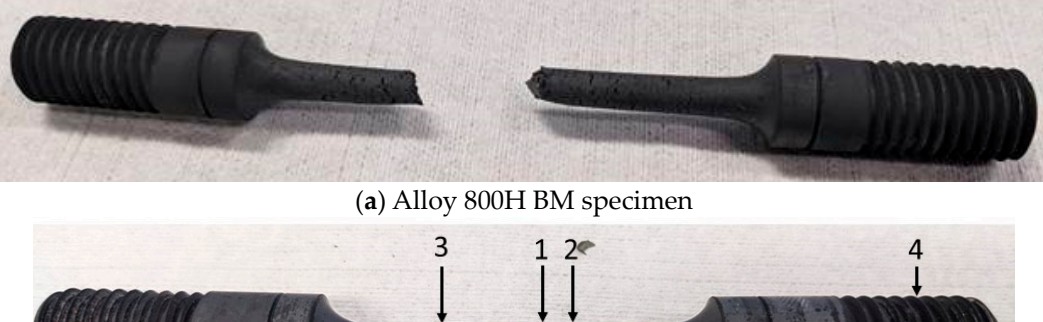

(**a**) Alloy 800H BM specimen

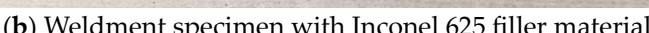

(**b**) Weldment specimen with Inconel 625 filler material

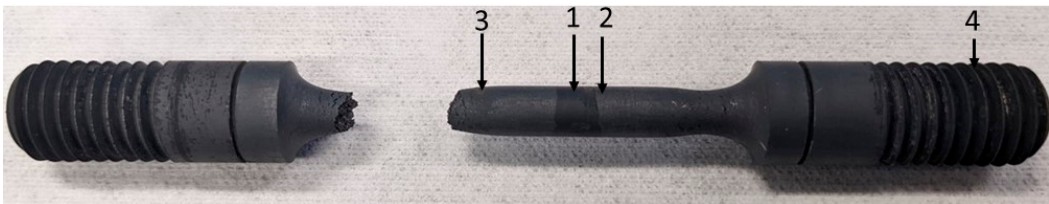

(**c**) Weldment specimen with Haynes 230 filler material

**Figure 6.** Images of the ruptured creep specimens. (**a**) Base metal specimen. (**b,c**) Weldment specimens with Inconel 625 and Haynes 230 filler materials after creep at 760 °C and 80 MPa. In (**b,c**), the regions for microstructure examinations are labeled as 1, FZ; 2, HAZ; 3, base metal close to facture surface adjacent to the fusion boundary; and 4, un-deformed base metal.

### 3.4. Microstructures

3.4.1. As-Received Alloy 800H Base Metal

The as-received 800H base metal has equiaxial grain structure with a number of twins. Figure 7a provides an optical image taken from the as-received BM. Second-phase particles, mainly Ti-rich Ti(C, N) in the interior of grains and Cr-rich carbides decorating grain boundaries, are distributed in the material. The as-received material contains dislocations dominantly located on (111) planes, and some dislocation cross-slips are also observed among the (111) slip planes, as shown in Figure 7b.

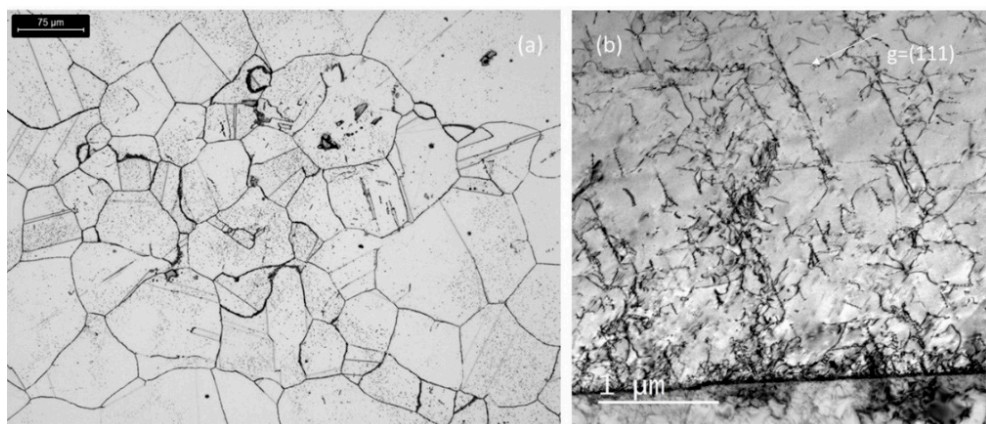

**Figure 7.** Optical (**a**) and TEM (**b**) images of as-received Alloy 800H BM.

### 3.4.2. As-Welded Microstructure

For the Inconel 625 filler material, Figure 8 shows an inverse pole figure map (IPFM) image of the FZ and HAZ (a) and a high-angle annular dark field (HAADF) image of the FZ (b). This image shows that second-phase particles were distributed in the grain interiors and at the grain boundaries of the FZ (Figure 8b). No significant grain growth was observed in the HAZ. The IPFM of the Haynes 230 filler weldment exhibited a similar characteristic, so it is not shown here.

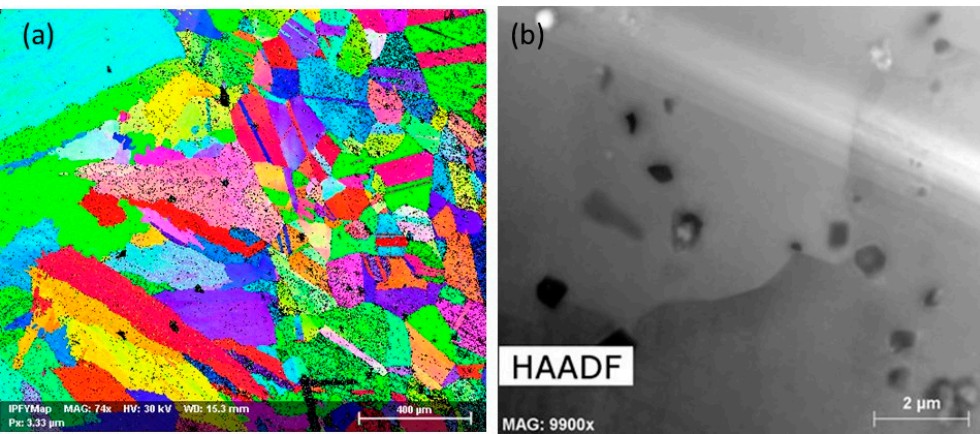

**Figure 8.** (**a**) IPFM image at as-welded condition of the weldment with Inconel 625 filler material, showing the FZ (**left**) and HAZ (**right**) grain morphology. (**b**) HAADF image showing the precipitates in the FZ.

In the FZ of the Inconel 625 filler material, the two main types of second-phase precipitates (SPPs) are Nb-, Ti-, and Si-rich MC carbide (face-centered cubic, or FCC) and Mo- and Si-rich intermetallic Laves phase (hexagonal close-packed, or HCP). For example, in Figure 9a, the SPPs at the grain boundary are an HCP Laves phase, verified by tilting them to different zone axes. Figure 9b shows the NBED pattern taken from the Laves phase at the zone axis of [0001]. The SSP in the grain interior is an MC carbide; Figure 9c shows the NBED from its [111] zone axis. Figure 9d–i provide the element EDX mapping of the studied region showing the difference in chemical composition between the two phases. Nano-sized MC carbide was also observed within the FZ. Figure 10 provides the HAADF image (a) and the element EDX mapping (b–f) of the region containing these nano-sized Nb- and Ti-rich MC precipitates. These carbides were embedded in a dense dislocation network, as shown in Figure 10a. The formation of dislocations likely occurred during the precipitation of MC, potentially contributing to the strengthening of the fusion zone for improved high-temperature properties. When comparing the SPPs in the FZ of the Inconel 625 filler weldment with those in the Haynes 230 filler weldment, it was observed that the majority of the SPPs in the latter are Ti- and Cr-rich carbides (Figure 11a–d) and W-Cr-Mn-Si-rich carbides (Figure 11e–h), and some of them are at nano-size scales (less than 100 nm). This indicates a distinct composition of precipitates in the two different weldments.

Figure 12 shows the microstructure of the as-welded HAZ adjacent to the fusion boundary; it appeared to be quite similar to that of the as-received material. The only noticeable difference is the presence of a thin film layer of Cr-rich carbides along the grain boundary, as illustrated in Figure 12c. This thin layer is likely a consequence of the thermal cycles during the welding process.

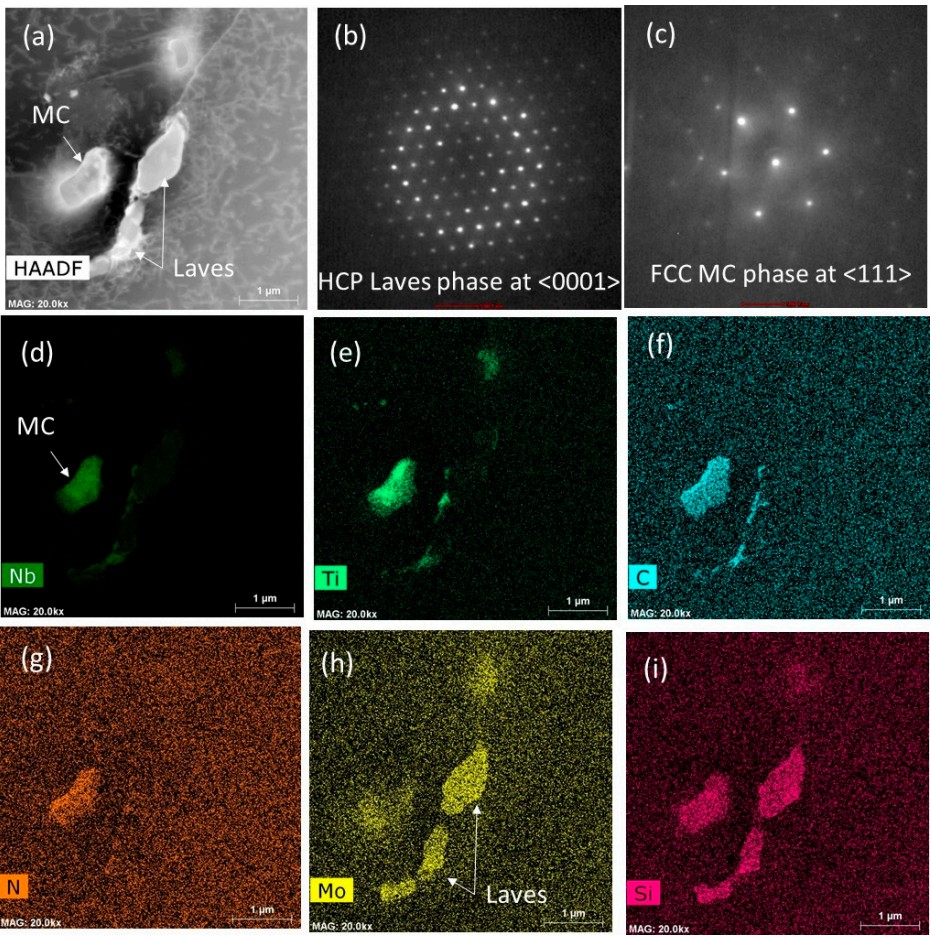

**Figure 9.** As-welded FZ of the weldment with Inconel 625 filler material. (**a**) HAADF image. (**b**,**c**) NBED patterns of the Laves and MC phases. (**d**–**i**) ChemiSTEM EDX maps of the studied region in (**a**).

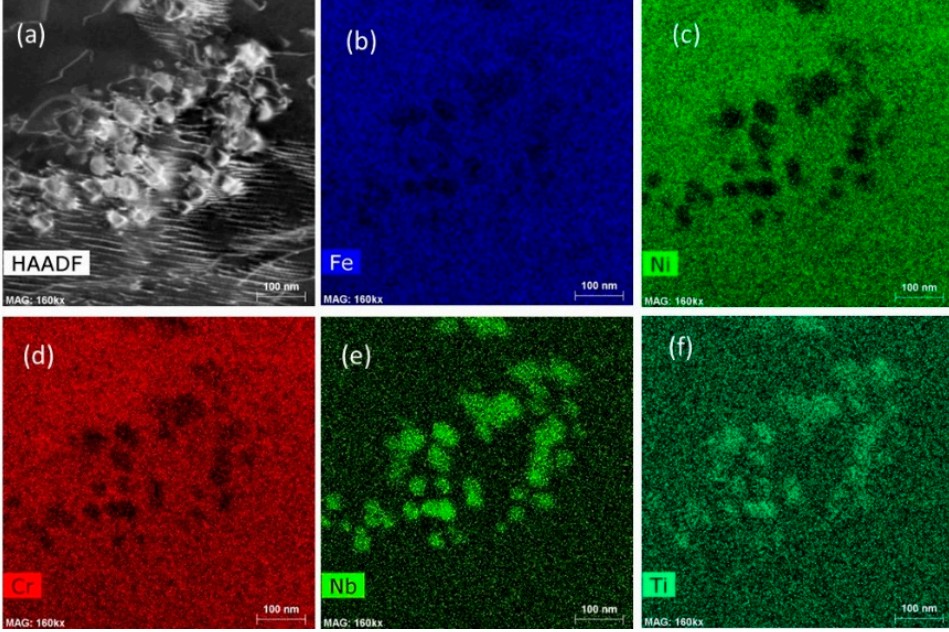

**Figure 10.** As-welded FZ using Inconel 625 filler material. (**a**) HAADF image. (**b**–**f**) ChemiSTEM EDX maps of Fe (**b**) Ni (**c**), Cr (**d**), Nb (**e**), and Ti (**f**) showing the Nb- and Ti-rich nano-sized MC carbides.

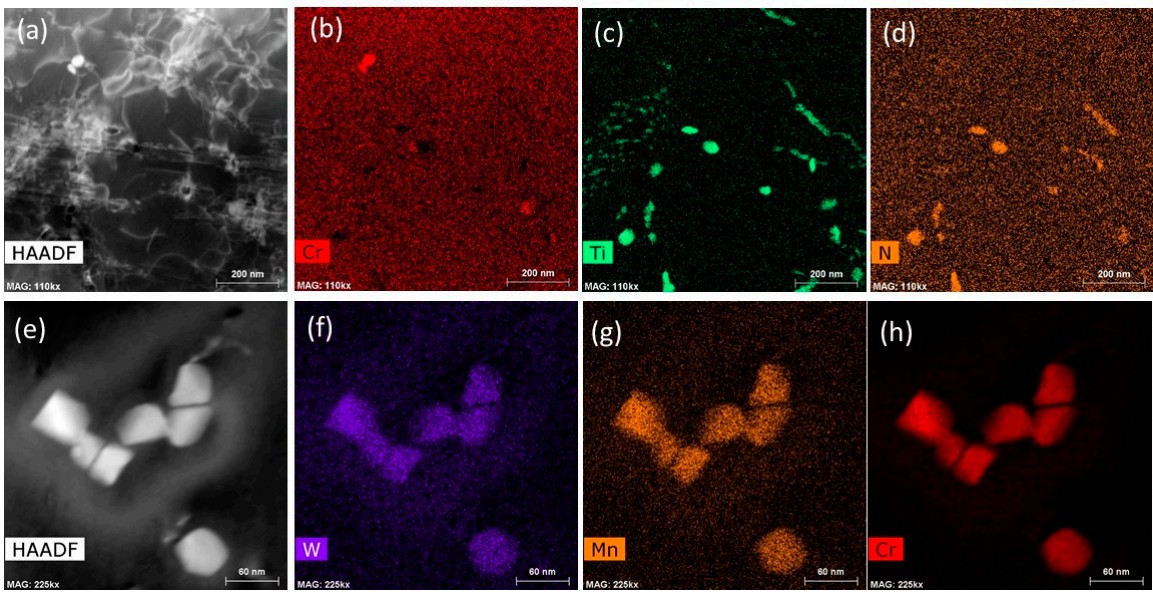

**Figure 11.** As-welded FZ using Haynes 230 filler material showing the different carbides. (**a**,**e**) HAADF images. (**b**–**d**) ChemiSTEM EDX maps of Cr, Ti, and N indicating MC (Ti-rich) and $M_{23}C_6$ (Cr-rich) carbides. (**f**–**h**) ChemiSTEM EDX maps showing nano-sized W-, Mn-, and Cr-rich carbides. Note that their type was not identified because of the nano size.

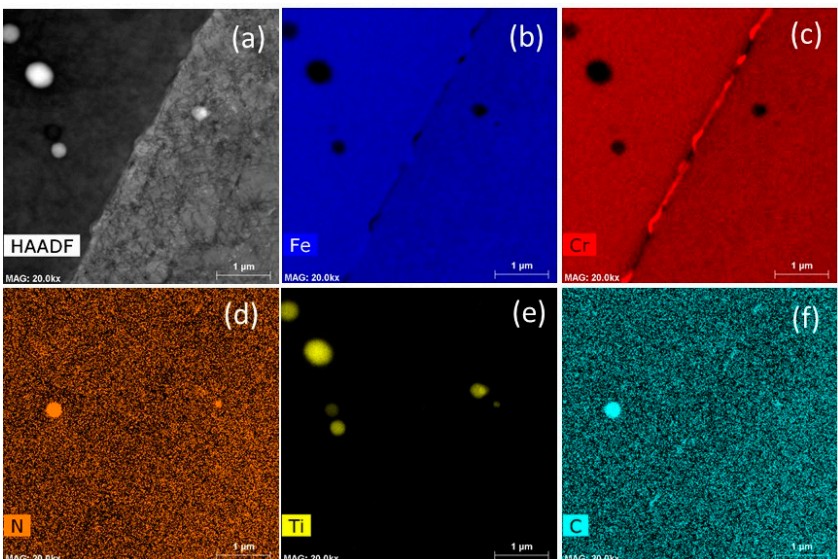

**Figure 12.** As-welded HAZ. (**a**) HAADF image. (**b**–**f**) ChemiSTEM EDX maps of Fe (**b**), Cr (**c**), N (**d**), Ti (**e**), and C (**f**).

### 3.4.3. Microstructure after Creep Rupture

FZ of Inconel 625 and Haynes 230 Filler Specimens (Region 1 in Figure 6)

After the creep deformation and high-temperature exposure, a significant amount of rod-shaped precipitations appeared in the FZ of the Inconel 625 filler weldment specimen, as shown in Figure 13a. The NBED pattern in Figure 13b indicated that it was the orthorhombic delta (δ) phase. The δ phase was enriched in Ni and Nb. Ni-, Nb-, and Ti-rich Laves particles, Si- and Mo-rich $M_6C$, Cr- and Mn-rich $M_{23}C_6$, and Ti-rich MC carbides were also observed. Figure 13c–e provides the NBED patterns of the observed Laves, $M_{23}Cr_6$, and $M_6C$ phases; their sizes varied, reaching a maximum of 2 μm. In contrast, the MC was smaller at less than 0.2 μm (Figure 13j).

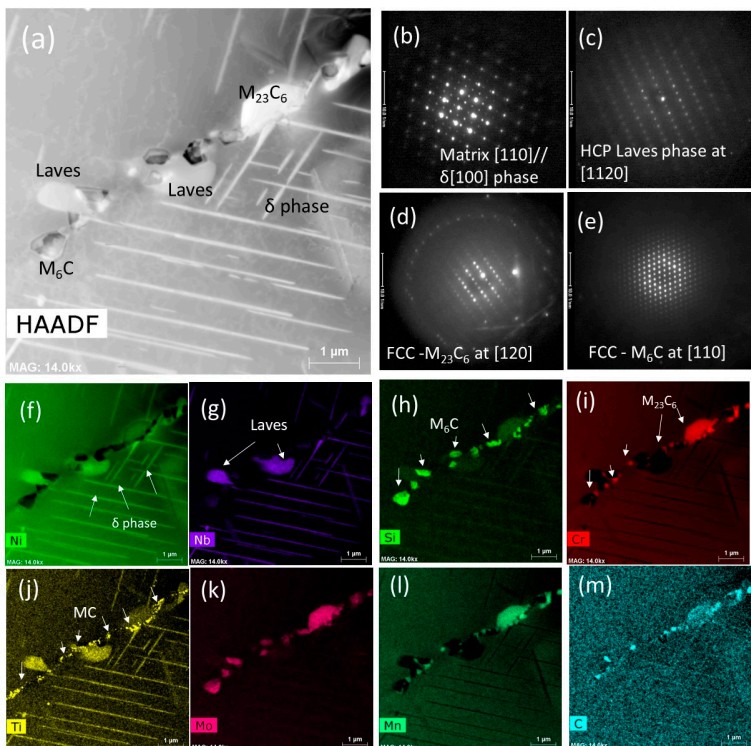

**Figure 13.** FZ of Inconel 625 filler specimen after creep rupture (1091 h) at 760 °C and 80 MPa. (**a**) HAADF image showing the identified SSPs. (**b**–**e**) NBED patterns taken from the SSPs and matrix in (**a**): rod-shaped δ phase at the zone axis of [100] and the parallel [110] zone axis of the matrix (**b**), HCP Laves phase at [1120] (**c**), $M_{23}C_6$ carbide at [120] (**d**), $M_6C$ at [110] (**e**), and ChemiSTEM element EDX maps showing chemical compositions of the SSPs (**f**–**m**).

In the FZ of the Haynes 230 filler material WM specimen, significant amounts of SSPs were also observed in the microstructure but no δ or Laves phase was observed. The principal precipitations are carbide phases. Figure 14 shows an HAADF image (a), NBED patterns (b,c), and ChemiSTEM element EDX maps (d–l). Figure 14d–l show Cr-rich $M_{23}C_6$ carbides (d), Ti(C, N) (e), and W-, Mo-, Nb-, Si-, Ti-, and Al-rich $M_6C$ carbide (f–j).

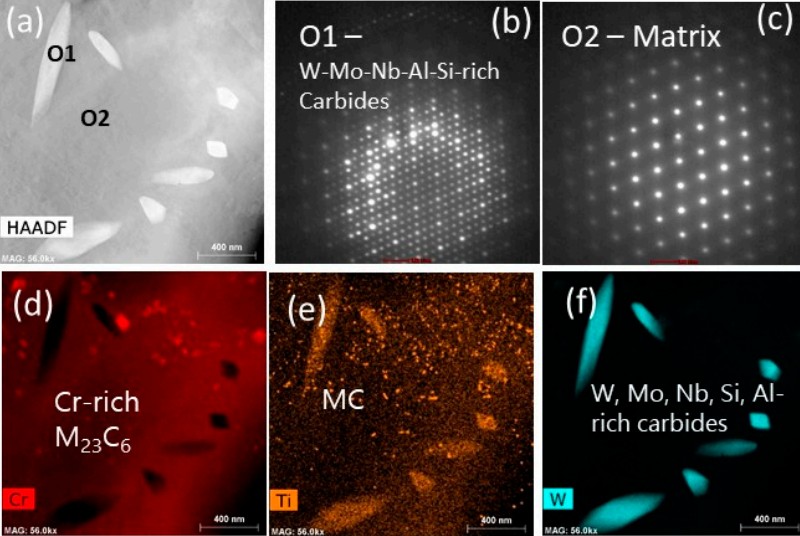

**Figure 14.** *Cont.*

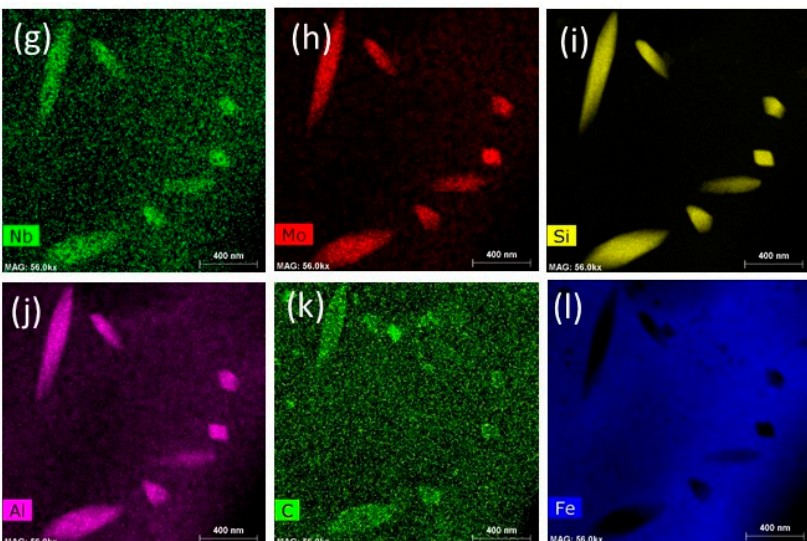

**Figure 14.** FZ of Haynes 230 filler weldment after creep rupture for 1643 h at 760 °C and 80 MPa. (**a**) HAADF image. (**b**,**c**) NBED patterns taken at [110] zone axis from the $M_6C$ SSP (O1) and matrix (O2). (**d**–**l**) ChemiSTEM element EDX maps.

HAZ Adjacent to Fusion Boundary (Region 2 in Figure 6)

In the HAZ, the Ti(C, N) and Cr- and Mn-rich $M_{23}C_6$ particles continued to precipitate during the creep test. Figure 15 provides the TEM bright field (a) and dark field (b,c) images and the electron diffraction patterns (d) showing the Ti(C, N) and $M_{23}C_6$ carbides. Figure 15d shows the electron diffraction pattern taken close to [112] zone axis of the matrix and the reflections from Ti(C, N) and $M_{23}C_6$. Note that the white arrowed diffraction spots are used to take the dark field images to show the FCC Ti(C, N) and FCC $M_{23}C_6$ in Figure 15b,c. It can be seen that grain boundaries are the preferred locations for both Ti(C, N) and $M_{23}C_6$ carbides. The Ti(C, N) carbides are smaller than the one of $M_{23}C_6$, and they are distributed throughout the grain interiors (Figure 15b). The precipitated $M_{23}C_6$ tend to be distributed adjacent to grain boundary and their amount gradually decreases towards the grain interiors (Figure 15c).

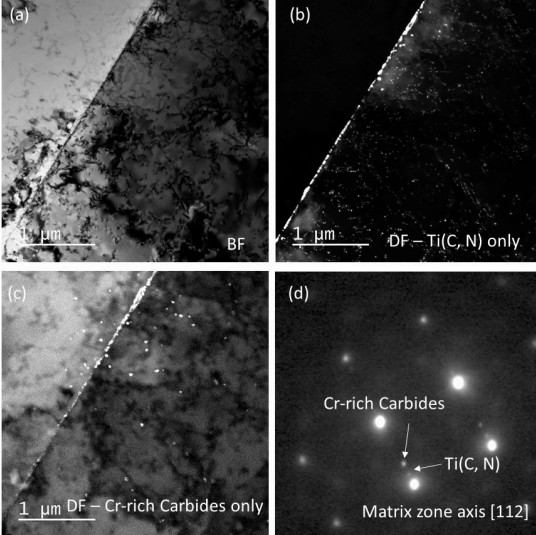

**Figure 15.** TEM images showing the Ti(C, N) and $M_{23}C_6$ carbides in the HAZ adjacent to the fusion boundary. (**a**) Bright field (BF) image. (**b**) Dark field (DF) image showing Ti(C, N) precipitates only. (**c**) DF image showing $M_{23}C_6$ only in the microstructure. (**d**) Diffraction pattern taken at the [112] zone axis showing the reflections from Ti(C, N) and $M_{23}C_6$, which were used to take the DF images in (**b**,**c**).

Base Metal Close to Rupture Surface (Region 3 in Figure 6)

Figure 16 provides the TEM bright field images taken from the base metal region close to the fracture surface at three different reflections, namely, g = $11\bar{1}$ (a), g = $1\bar{1}1$ (b), and g = 111 (c). It can be seen that the dislocation structure in the form of dislocation loops is well aligned with the <110> {111} slip systems. These dislocation loops were produced due to the interaction between dislocation and precipitates via the Orowan dislocation-precipitate bypass mechanism [17]. Cross slips were also observed to take place, indicating that screw dislocations move from one slip plane to other {111} planes during creep deformation. The bypassed precipitates in the {111} slip planes were identified as Ti(C, N) with a size of about 10 nm.

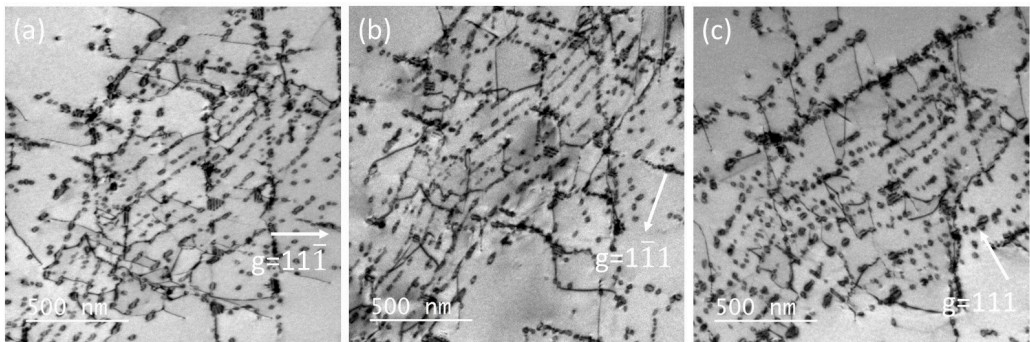

**Figure 16.** TEM BF images showing the Orowan dislocation loops and the cross slips, taken at three different reflections. (**a**) g = $11\bar{1}$. (**b**) g = $1\bar{1}1$. (**c**) g = 111.

Figure 17 shows the distribution and morphology of $M_{23}C_6$ and Ti(C, N) precipitates in the deformed Alloy 800H base metal section of the weldment specimen. They appeared in three main manners. The first manner was block-shaped $M_{23}C_6$ (arrowed in Figure 17a) segregated at the grain boundary and having a size of hundreds of nanometers. The second was cube-shaped $M_{23}C_6$ clustered adjacent to the grain boundary and also precipitated along the slip planes or dislocations. These had a size of about 50 nm and contained Moiré fringes, as shown in the enlarged area in Figure 17b. The amount of these $M_{23}C_6$ gradually decreased towards the grain interiors. The third was cube-shaped $M_{23}C_6$ clusters that consisted of large blocky $M_{23}C_6$ (hundreds of nanometers in size) and smaller ones (about 50 nm in size and containing Moiré fringes), as shown in Figure 17c. Figure 17d provides an enlarged TEM BF image showing small $M_{23}C_6$ about 50 nm in size (arrowed in (d)) containing Moiré fringes and Orowan dislocation loops with Ti(C, N) precipitates.

High-Temperature Ageing Structure in Incoloy 800H (Region 4 in Figure 6)

Figure 18 reveals the microstructure of the un-deformed base metal. Cr-rich $M_{23}C_6$ with Moiré fringes and nano-sized Ti-rich Ti(C, N) carbides were the two predominant precipitates. Both types of precipitates tended to appear along the pre-existing dislocation within the microstructure. The absence of an Orowan dislocation loop surrounding the Ti(C, N) indicated that these carbides precipitated as a result of thermal ageing without interaction with the dislocation movement. This is in contrast to the observation in the gauge section presented in Figures 16 and 17d.

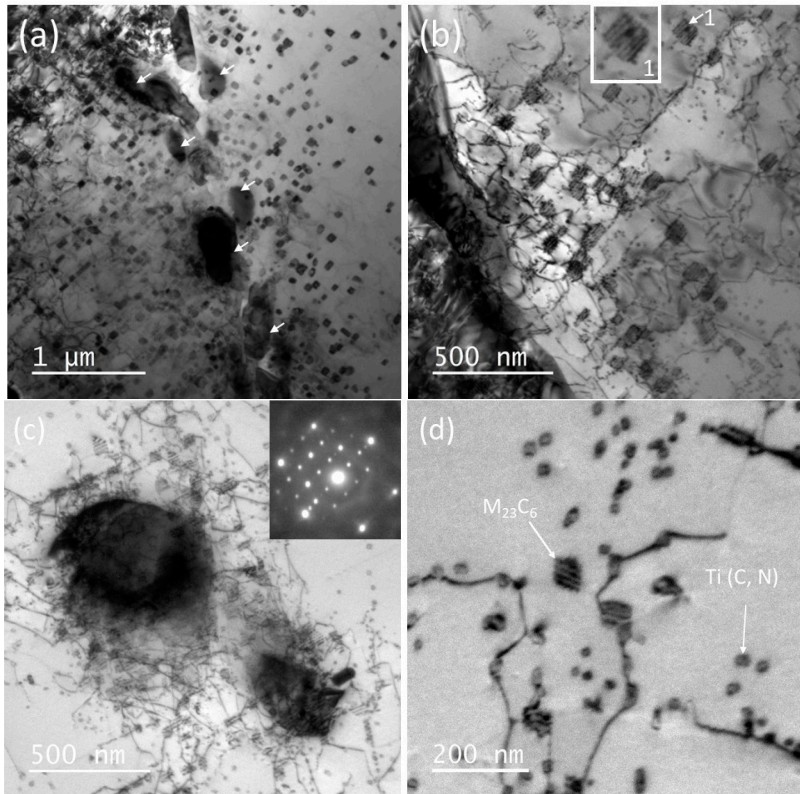

**Figure 17.** (**a**–**d**) TEM BF images of base metal in the necked region close to the fracture surface (**a**), $M_{23}C_6$ near grain boundary (**b**), $M_{23}C_6$ clusters with dislocation networks (**c**), and small $M_{23}C_6$ containing Moiré fringes and Ti(C, N) precipitates with Orowan dislocation loops (**d**). Note that the arrows show the block-shaped $M_{23}C_6$ segregated at the grain boundary.

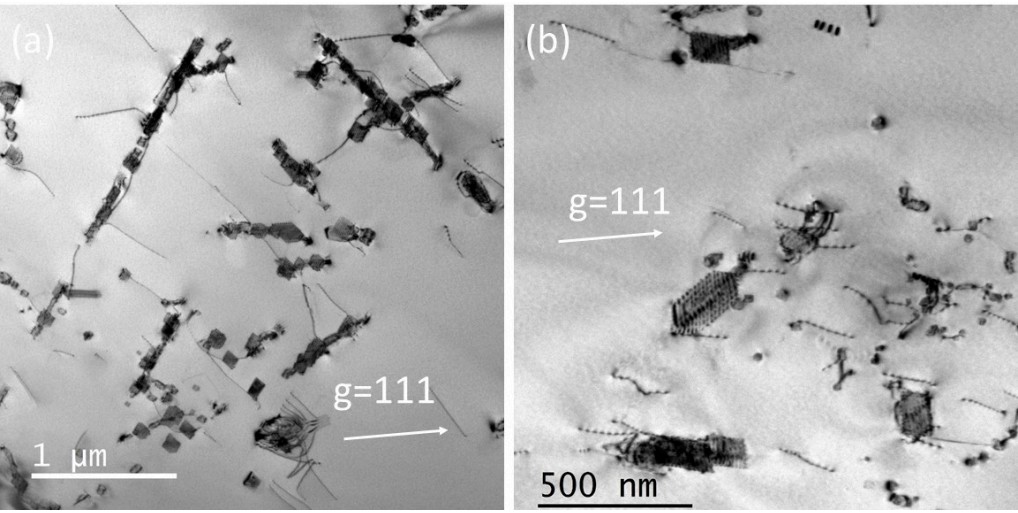

**Figure 18.** TEM BF images in the un-deformed region, (**a**) precipitates (**b**) $M_{23}C_6$ showing with Moiré fringes and nano-sized Ti(C, N).

Fractography

Figures 19–21 provide SEM micrographs of the fracture surface from the Alloy 800H BM specimens and the weldment specimens with Inconel 625 and Haynes 230 filler materials. The SEM micrographs reveal obvious surface oxidation on the fracture surface of all specimens due to prolonged exposure in air during the tests. The surface oxidation appears to be more significant in the weldment specimens compared with the BM specimens. This

could be attributed to the longer rupture time, i.e., 1091 h and 1643 h with the Inconel 625 and Haynes 230 filler metal specimens as opposed to 467 h with the BM specimen. The fracture surface of the Alloy 800H BM specimen showed predominantly transgranular facture failure (Figure 19a,b) and contained a few secondary cracks, as arrowed in Figure 19c. The fracture surface of the weldment specimens displayed a combination of transgranular rupture and intergranular rupture. Arrows in Figures 20c and 21c indicate the presence of intergranular cracks on the fracture surface of the weldment specimens.

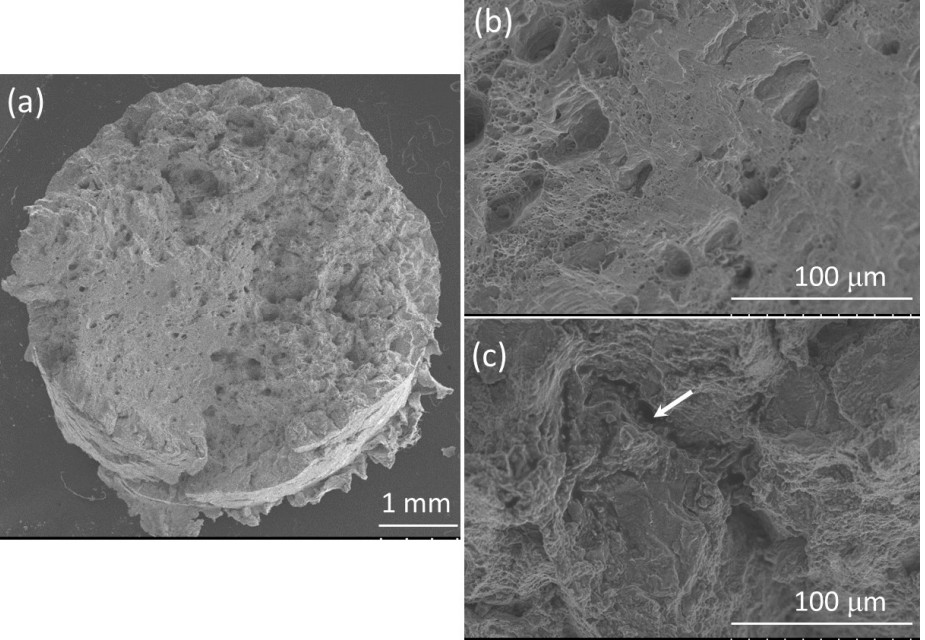

**Figure 19.** (**a**) SEM micrograph of fracture surface in Incoloy 800H BM specimen, (**b**,**c**) fracture micrograph with a higher magnification. The arrow in (**c**) indicates a secondary crack.

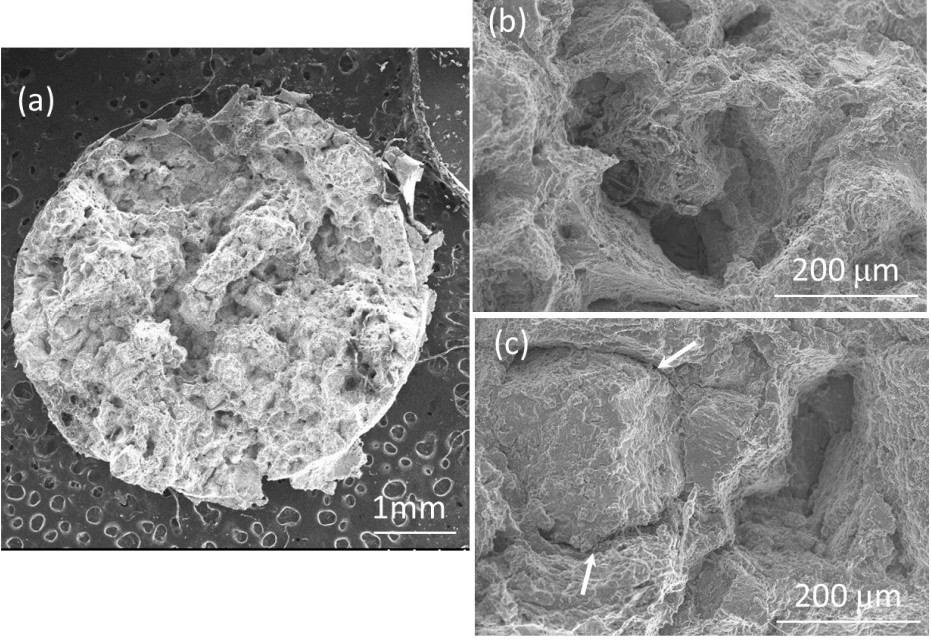

**Figure 20.** (**a**) SEM micrograph of the fracture surface in the WM specimen with the Inconel 625 filler material, (**b**,**c**) fracture micrograph with a higher magnification. The arrows in (**c**) indicate the intergranular cracks.

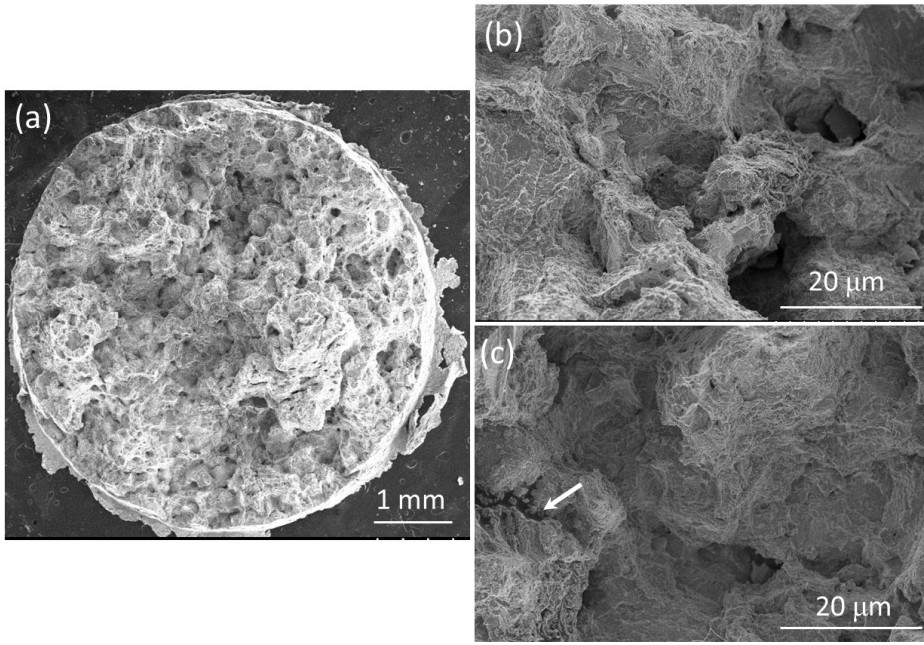

**Figure 21.** (**a**) SEM micrograph of fracture surface in the WM specimen with the Haynes 230 filler material, (**b**,**c**) fracture micrograph with a higher magnification. The arrow in (**c**) indicates an intergranular crack.

## 4. Discussion

### 4.1. High-Temperature Creep Deformation and Mechanisms

The Alloy 800H weldments with Inconel 625 filler and Haynes 230 filler materials exhibit a strengthened property in both high-temperature tensile and creep behaviors. This is mainly because of the higher hardness and apparent YS in the gauge section, which consists mainly of FZ and HAZ with a small volume faction of the unaffected base metal. The higher apparent 0.2% YS in the weldment caused a lower ratio of the applied stress to the apparent yield stress, about 0.40 versus 0.68, in the weldment specimens compared with the base metal specimen when the same stress of 80 MPa was applied in the current study.

In the deformation mechanism maps developed by Beardsley et al. [18], three creep mechanisms were presented: low-temperature power-law, high-temperature power-law, and Coble creep in Alloy 800H with an average grain size of 100 μm, at test temperatures of 750 to 1020 °C and test stresses of 14.1 to 105 MPa. In this study, high-temperature power-law dislocation creep was likely to be the dominant mechanism in the Alloy 800H base metal (HAZ and unaffected base metal in the gauge section) under the current test conditions (stress and temperature). Moreover, the dislocation bypass on the precipitates by bowing around them (Orowan mechanism) or cutting through them (shearing mechanism) was an important contributor to facilitate dislocation slips in addition to the dislocation climb.

The physical-based model based on the climb and Orowan mechanisms has been widely developed for creep rate and lifetime prediction [19–21]. Li et al. [22] developed a high-temperature creep model by considering the coexistence of climb, cutting, and Orowan mechanisms during creep processes, and the contribution of these mechanisms were considered to be dependent on the precipitate size and distribution relative to the slip plane. The considerable amounts of Orowan dislocation loops and cross slips in the Alloy 800H base metal material of the weldment specimens have provided strong evidence of the dislocation power-law creep and of the coexistence of climb, cutting, and Orowan mechanisms under the current test conditions.

In the current study, both the base metal and the weldment specimens show a significantly longer tertiary creep stage than the primary and secondary stages, as shown in Figure 5. In particular, the secondary creep stage is relatively short. It took only 60 h (13%

of the total time) in the primary and secondary stages, and 407 h in the tertiary stage until final rupture in the base metal material. In comparison, the weldment specimen took about 30% of the total time to reach the tertiary stage. Tertiary creep is a stage characterized by an increased creep rate due to a high recovery rate. It is often associated with metallurgical changes, including coarsening of precipitate particles and formation of voids, ultimately leading to failure [23]. It is likely that the competitive process between the significant precipitation and growth of Ti(C, N) and $M_{23}C_6$ played a role in the prolonged tertiary creep stage in the studied 800H and its weldment. On the one hand, the precipitation and growth of these carbides could have increased the creep resistance as obstacles of dislocation movement; on the other hand, it also caused the initiation and growth of the voids due to the growth of these precipitates, which will promote voids, cavities, and cracking especially at the grain boundaries. The latter process is aggressively more prevalent than the former with increasing creep time, so that the creep strain rate increases gradually at the tertiary stage.

According to the time–temperature–transformation (TTT) diagram of Incoloy 800 in [24], carbides continue to precipitate after prolonged exposure at high temperature. It was reported that the fine carbides (Ti and Cr carbides) are precipitated at 550 to 760 °C in regions of high density of dislocations inside the grains after short periods of thermal exposure, leading to strengthening of the grains. At the grain boundaries, coarse chromium carbides were produced, leading to chromium depletion in their vicinity so that precipitation-free zones (PFZs) caused low strength beside the grain boundaries [24]. In this study, Cr depletion and PFZs were not observed in this region; however, significant amounts of coarse Cr- and Mn-rich $M_{23}C_6$ carbides were present along the grain boundaries, as shown in Figure 17a.

In the final stage of the tertiary phase, the nucleation of cavities followed by growth and interlinkage along the grain boundary led to final failure [25]. As observed in both base metal and weldment specimens, cracks developed along a direction perpendicular to the applied load, and intergranular cracking was the more dominant fracture mode in the weldment specimens due to the prolonged high-temperature exposure and extensive carbide precipitates and growth (Figures 19–21). Intergranular cracking is considered to be the dominant fracture mechanism of high-temperature Incoloys [25]. It was reported that creep cavities and intergranular cracks developed in the perpendicular direction of the stress in Inconel 617, and $M_{23}C_6$ carbides were found to form along the grain boundary and inside the matrix at temperatures of 649 to 1093 °C [26].

### 4.2. Precipitate Evolution during High-Temperature Creep

The precipitates of the studied materials exhibited significant evolution and changes after long-term exposure at high temperature and stress. Carbides (MC or Ti(C, N), $M_{23}C_6$, and $M_6C$) have been commonly observed in the microstructure of FZ, HAZ, and base metals, with significant changes in their amount and morphology evident across different conditions: as-received, as-welded, and after creep. The thermal exposure and creep load led to a substantial change in the compositions and increase in morphological variations of carbides.

The study reveals significant changes in the types and compositions of precipitates in the FZ of weldment specimens after creep tests. In both as-weld and post-creep conditions, the FZ of the Inconel 625 filler weldment exhibited an intermetallic Laves phase. A Laves phase is a hexagonal close-packed intermetallic compound with an $AB_2$-type structure. The 'A' atoms, such as Fe, Ni, or Cr, and 'B' atoms, including Mo, Nb, or Si, play crucial roles in the formation of a Laves phase [27]. The Laves phase forms in Inconel 625 during solidification such as in a welding process, because the carbon content of Inconel 625 is generally sufficiently high to promote the formation of both carbides and intermetallics at the end of solidification [27–29]. The Laves phase also continues to precipitate during the prolonged heat treatment in Inconel 625 according to the TTT diagram [27]. Laves phases are extremely hard and not plastically deformable at room temperature. Therefore, the

Laves phase has been considered to have a detrimental effect on the hardening of a material and encourages brittleness.

In addition, a δ phase was observed after the creep deformation in the FZ of the Inconel 625 filler WM specimen. The δ phase initiates precipitation after approximately 20 h of heating within the temperature range of 650–980 °C in Inconel 625 [27]. The δ phase is Nb- and Ni-enriched and has an acicular morphology that provides a distinctive needle-like or rod shape. The Laves particles are very similar in morphology to the blocky, irregularly shaped MC and $M_{23}C_6$ carbides. Like the Laves phase, δ particles formed during creep are also detrimental and reduce the ductility and toughness of the material. They can cause stress concentration and become a favorable location for the initiation and propagation of cracks. The solid solution strengthening can be significantly weakened, because he precipitations of these phases could deplete the solute atoms that are contributing in solid solution hardening and, therefore, reduce the strength of materials; corrosion resistance can also be reduced as a result of the solid solution depletion [27].

In the current study, whereas the formation of δ and Laves phases in the Inconel 625 filler material potentially introduces brittleness in the fusion zone, their effect on creep resistance is mitigated in comparison to the less creep-resistant Incoloy 800H base metal. The failure, therefore, occurs in the base metal rather than in the fusion zone. In contrast, the weld metal in the fusion zone of Haynes 230 did not exhibit deleterious δ and Laves phases. Principal phases precipitated from solid solution were of the same types before and after creep. They were all carbides, i.e., Ti(C, N), Cr- and Mn-rich $M_{23}C_6$ carbides, and W-, Mo-, Nb-, Si-, Ti-, and Al-rich $M_6C$ carbides. The observation that the weldment with Haynes 230 filler material maintained the same types of precipitates before and after creep, particularly carbides, suggests better phase stability compared to the Inconel 625 filler material.

## 5. Conclusions

In this study, high-temperature creep properties at 760 °C and 80 MPa of Alloy 800H and its weldments using Inconel 625 and Haynes 230 filler materials were investigated and compared. The microstructures before and after the creep test and the fracture mode were reported. Creep mechanisms, dislocation structure, and precipitates evolution were explicated. The following conclusion can be drawn:

- High-temperature apparent tensile yield strength and creep resistance of Incoloy 800H welds at 80 MPa and 760 °C were significantly enhanced by the addition of Inconel 625 and Haynes 230 into the Alloy 800H weldments. Both of the weldments showed longer creep rupture time but lower rupture strain compared with the BM specimen.
- Significant dislocation slip and interaction with precipitates were observed in the microstructure, indicating a high-temperature power-law creep mechanism. Dislocation bypassing through the Orowan mechanism accompanied by climb and cutting facilitated dislocation slips during high-temperature creep deformation.
- Microstructural characterization revealed that extensive precipitation took place after the prolonged creep testing at high temperature. A large number of sub-micron-sized carbides (MC and $M_{23}C_6$) were observed in the microstructure of FZ, HAZ adjacent to the fusion boundary, and base metals under various conditions (as-received, as-welded, and creep-tested). The varied sizes and locations of the $M_{23}C_6$ and MC carbides suggest a complex microstructural evolution during the creep test.
- The weldments with Inconel 625 filler material exhibited detrimental δ and Laves phases in the weld metal after the creep test. Although the failure occurred in the base metal rather than in the fusion zone under the current test conditions, the presence of these phases could cause potential crack initiation after prolonged high-temperature ageing.
- The weldment with Haynes 230 filler material demonstrated superior phase stability and improved creep rupture properties compared to the one with Inconel 625 filler

material. This suggests that Haynes 230 could be a promising filler material for further investigations into Alloy 800H applications.

In conclusion, microstructural evolutions and phase stability of filler materials and base metals are important aspects in high-temperature creep deformation. They should be evaluated and understood when studying the high-temperature materials and their weldments in high-temperature reactor applications.

**Author Contributions:** W.L.: conceptualization, investigation, methodology, funding acquisition, writing—original draft; L.X., L.W. and Q.D.: methodology, validation, writing—review and editing; M.I. and R.S.: writing—review and editing. All authors have read and agreed to the published version of the manuscript.

**Funding:** This work was supported by Atomic Energy of Canada Limited's Federal Nuclear Science & Technology Work Plan.

**Informed Consent Statement:** Not applicable.

**Data Availability Statement:** The data presented in this study are available on request from the corresponding author.

**Conflicts of Interest:** The authors declare no conflicts of interest.

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
