# Peer review of "High-Temperature Creep and Microstructure Evolution of Alloy 800H Weldments with Inconel 625 and Haynes 230 Filler Materials"

_applsci, doi:10.3390/app14041347_

Round 1
Reviewer 1 Report
Comments and Suggestions for Authors
This paper compares the high Temperature creep performance and microstructure evolution in Ally 800H and its weldments with Inconel 625 and Haynes 230 filler materials tested at 760 oC under a stress of 80 MPa. The weld specimens contain FZ/HAZ in the gauge section, as shown in Fig. 2. , so rigorously speaking, the total elongation is e = fBM eBM + fFZ/HAZ eFZ/HAZ, where fBM and fFZ/HAZ are volume fraction of the BM and FZ/HAZ, respectively. Apparently, because the filler materials are stronger than the BM, the overall creep elongation of the weldments was smaller and the fracture still occurred in the BM section. The authors should try to evaluate the true strain of FZ/HAZ, according to the above composite rule.
If the authors only concerns the overall performance, what's the purpose it serves by conducting detailed microstructural examinations such as IPFM, HAADF NBED ChemiSTEM EDX on FZ in this study? The reviewer believes it clarifies the microstructural effect only once the true creep strain of FZ is evaluated. Haven't IN625 and Haynes 230 weld microstructures been studied in the literature, which are widely used materials?
In the discussion, it is said that the formation of delta phase and Laves phase in FZ are detrimental, then how the effect is "mitigated" to the BM region, as it is implied in the sentence "the mitigated effect on creep resistance is evident in comparison to the less creep-resistant Incoloy 800H base metal."?
The authors quoted a previous study on deformation mechanism map of Alloy 800, which recognize LT creep (presumably dominated by intragranular dislocation glide), HT creep (presumably controlled by intragranular dislocation creep) and Coble creep. Please comment on what mechanism categories the authors tests fell into? And what mechanism is responsible for the dominant intergranular fracture observed in the authors study? The authors missed discussion of grain boundary sliding as a possible deformation and failure mechanism. The reviewer thinks the fractographic evidence points to that.
Comments on the Quality of English Language- Change "oC" to oC.
- Correct "Error! Reference source not found".
- p16, first line, "tyoe" -> type.
- p. 18, the sentence "The physical-based model because of the climb and Orowan mechanisms was widely developed for the prediction of the creep rate and its correlated creep lifetime [19]-[21]." is incomplete.
Author Response
We thank the reviewers for the comments and appreciate the feedback and the time spent on our manuscript. We found the comments very helpful and carefully addressed all of them in the revised manuscript. The modified/added parts in the manuscript are highlighted in red. We hope that our comprehensive response below is satisfactory.
Comments and Suggestions for Authors
This paper compares the high Temperature creep performance and microstructure evolution in Ally 800H and its weldments with Inconel 625 and Haynes 230 filler materials tested at 760 oC under a stress of 80 MPa. The weld specimens contain FZ/HAZ in the gauge section, as shown in Fig. 2. , so rigorously speaking, the total elongation is e = fBM eBM + fFZ/HAZ eFZ/HAZ, where fBM and fFZ/HAZ are volume fraction of the BM and FZ/HAZ, respectively. Apparently, because the filler materials are stronger than the BM, the overall creep elongation of the weldments was smaller and the fracture still occurred in the BM section. The authors should try to evaluate the true strain of FZ/HAZ, according to the above composite rule.
- Thanks for the great comment. We recognized that the deformation in the weldments within gage length is heterogeneous due to the difference of microstructure, phase component, and then mechanical strength among FZ, HAZ and BM. Therefore, the measured strain is an average strain among BM, HAZ and FZ. One of our future work is to measure localized strain in the three zones using DIC or other option, and finite element simulation is also considered to quantitatively distinguish the plastic deformation behaviour among BM, HAZ and FZ and calculate the true strain of FZ/HAZ.
If the authors only concerns the overall performance, what's the purpose it serves by conducting detailed microstructural examinations such as IPFM, HAADF NBED ChemiSTEM EDX on FZ in this study? The reviewer believes it clarifies the microstructural effect only once the true creep strain of FZ is evaluated. Haven't IN625 and Haynes 230 weld microstructures been studied in the literature, which are widely used materials?
- One of the purposes to perform the detailed microstructural examinations on FZ in this study is to compare the microstructural evolution before and after creep. Based on the authors’ knowledge and literature review, microstructural evolutions of IN625 and Haynes as filler material in high-temperature creep tests are very limited. Our findings are to emphasize that microstructural evolutions and phase stability should be essential factors in evaluating high temperature materials and their weldments in high temperature reactor applications. As addressed in the previous comment, we will work on the localized strain measurement and analysis, and the microstructural effect will be an important input for such analysis. Thanks for the suggestion.
In the discussion, it is said that the formation of delta phase and Laves phase in FZ are detrimental, then how the effect is "mitigated" to the BM region, as it is implied in the sentence "the mitigated effect on creep resistance is evident in comparison to the less creep-resistant Incoloy 800H base metal."?
- Thanks for pointing out this inaccurate statement in the manuscript. The statement in the manuscript was not expressed precisely. It has been revised to “In the current study, while the formation of δ and Laves phases in the Inconel 625 filler material potentially introduces brittleness to the fusion zone, their effect on creep resistance is mitigated in comparison to the less creep-resistant Incoloy 800H base metal.”
The authors quoted a previous study on deformation mechanism map of Alloy 800, which recognize LT creep (presumably dominated by intragranular dislocation glide), HT creep (presumably controlled by intragranular dislocation creep) and Coble creep. Please comment on what mechanism categories the authors tests fell into? And what mechanism is responsible for the dominant intergranular fracture observed in the authors study? The authors missed discussion of grain boundary sliding as a possible deformation and failure mechanism. The reviewer thinks the fractographic evidence points to that.
- As described in the last sentence of Section 4.1, we consider the dislocation creep mechanism is the dominant at the current rest conditions. Because there are considerable amounts of Orowan dislocation loops and cross slips in the Alloy 800H base metal material of the weldment specimens. This showed strong evidence of the dislocation power-law creep, and the coexistence of climb, cutting, and Orowan mechanisms in the current test condition.
- Grain boundary sliding was not considered in the current test conditions. At high temperature deformation, metallurgical changes, including coarsening of precipitate particles and formation of voids, ultimately can lead to failure or fracture. In the present study, significant precipitation and growth of Ti(C, N) and M23C6 played a role in the final fracture. On one side, they can increase the creep resistance as obstacles of dislocation movement, on the other side, it also caused the initiation and growth of the voids due to the growth of these precipitates which will promote voids, cavities and cracking especially in the grain boundaries. The later process is aggressively more prevailing than the former one with the increasing creep time, so that the creep strain rate increases gradually in the tertiary stage. Therefore, the nucleation of cavities followed by growth and interlinkage along grain boundaries and the carbide precipitates in the grain boundary is considered to play a dominant role to the final fracture. These statements are in the second paragraph of Section 4.1.
Comments on the Quality of English Language
- Change "oC" to oC.
Corrected. Thanks
- Correct "Error! Reference source not found".
Corrected. Thanks
- p16, first line, "tyoe" -> type.
Corrected. Thanks
- p. 18, the sentence "The physical-based model because of the climb and Orowan mechanisms was widely developed for the prediction of the creep rate and its correlated creep lifetime [19]-[21]." is incomplete.
The sentence was revised as “The physical-based model based on the climb and Orowan mechanisms was widely developed for the creep rate and lifetime prediction [19]‑[21]”
Reviewer 2 Report
Comments and Suggestions for Authors
In this manuscript, Inconel 625 and Haynes 230 were chosen as filler materials to produce Alloy 800H weldments using the Gas Tungsten Arc Welding technique. The high-temperature tensile and creep properties of these weldments were reviewed. The microstructure for the creep test and fracture mode were discussed. There are minor comments need to be reassessed:
Line 18: time or times?
Line 28: Add a keyword 'weldment'
Line 73-75: Could the authors mention how many plates employed in this study?
Line 79-80: Error.. sentence means nothing and should be corrected?
Table 1: Why is INCO 625 & Haynes 230 given nominal compositions opposed to Alloy 800H? Could the authors also mention uncertainty for compositions and which combusion analysis was used for each element?
Figure 2, 3, 4, 5..: punctuation *.*
Line 102-103: Why did use a load 0.5 kg and not 1 kg more accurate?
Line 110: High temperature tensile/creep tests
Line 164-168: re-formulate this paragraph?
Line 193 :What is mean that?
Figure 5: Could the authors re-shape graphs to better visual and clear?
Figure 4 and 5: Could the authors verify these graphs with at least two experiments? So, it can build the results based on three measurements instead of one reading?
Table 2: Could the authors show uncertainty?
Line 225: How you know that annealing and not deformation twins? Provide citation?
Line 227: Double stop points?
Line 241-242: Need to write the whole sentence on the same line? There is a space to continue write on the same line?
Figure 11: How could you estimate phase analysis in alloy such as M23C6, M6C carbides and Ti (N, C)?
Line 282, 284, 285, and 286: Correct and write Figure/cross-reference?
Line 315: What is the morphology and shape of M23C6?
Line 317: Could the authors show where Moiré fringes located relating to grain boundary?
Line 321: Could you show where theses particles placed?
Line 327: Do you mean 'types' instead 'tyoes'?
Comments on the Quality of English Language
Please, se above our comments/feedbacks
Author Response
We thank the reviewers for the comments and appreciate the feedback and the time spent on our manuscript. We found the comments very helpful and carefully addressed all of them in the revised manuscript. The modified/added parts in the manuscript are highlighted in red. We hope that our comprehensive response below is satisfactory.
Comments and Suggestions for Authors
In this manuscript, Inconel 625 and Haynes 230 were chosen as filler materials to produce Alloy 800H weldments using the Gas Tungsten Arc Welding technique. The high-temperature tensile and creep properties of these weldments were reviewed. The microstructure for the creep test and fracture mode were discussed. There are minor comments need to be reassessed:
Line 18: time or times?
- Changed to times
Line 28: Add a keyword 'weldment'
- Thanks, added the keyword 'weldment'
Line 73-75: Could the authors mention how many plates employed in this study?
- Two to three plates were welded for each filler material. About 25 to 30 specimens can be machined from each plate.
Line 79-80: Error.. sentence means nothing and should be corrected?
- The “Error! Reference source not found” has been corrected. It should be “Table 1”
Table 1: Why is INCO 625 & Haynes 230 given nominal compositions opposed to Alloy 800H? Could the authors also mention uncertainty for compositions and which combusion analysis was used for each element?
- The compositions of as-received Alloy 800H was provided by the supplier and they were provided in Table 1. However, the compositions of INCO 625 & Haynes 230 filler metal and no combustion analysis were not measured, so their nominal composition were used in Table 1.
Figure 2, 3, 4, 5..: punctuation *.*
- Punctuation “.” After “Figures 2-5” was added. Thanks.
Line 102-103: Why did use a load 0.5 kg and not 1 kg more accurate?
- We do not quite understand the reviewer meaning on this comment.
Line 110: High temperature tensile/creep tests
- Corrected as “High temperature tensile and creep tests”.
Line 164-168: re-formulate this paragraph?
- The sentences were revised: “In the machined weldment specimens, as illustrated in Figure 2, a 25 mm gauge section consisted of approximately 6.4 mm weld metal in the middle, 7 mm HAZ and 1 to 2 mm of unaffected BM materials on each side of FZ, i.e., approximately 26% of volume fraction in the FZ, 60% in the HAZ, and 14% in the unaffected base metal materials, respectively. “
Line 193 :What is mean that?
- “Error! Reference source not found” has been corrected. It should be “Table 2”
Figure 5: Could the authors re-shape graphs to better visual and clear?
- The figures have been revised to make them easier to read.
Figure 4 and 5: Could the authors verify these graphs with at least two experiments? So, it can build the results based on three measurements instead of one reading?
- The plots in the figures are from one experiment. We did some repeating tests on tensile tests and they had good repeatability. For the creep test report in Figure 5, currently we only have one test performed for each material because of long duration of the creep tests.
Table 2: Could the authors show uncertainty?
- As addressed in the previous comment, one test was performed for each material for the current long term creep test.
Line 225: How you know that annealing and not deformation twins? Provide citation?
- You are right. The as-received 800H was in the solution annealed condition at 1167 oC for 32 mins, but the twins may not be totally annealing twins. The “annealing” was removed.
Line 227: Double stop points?
- Thanks.
Line 241-242: Need to write the whole sentence on the same line? There is a space to continue write on the same line?
- This seems to be an editing issue. They have been corrected. Thanks.
Figure 11: How could you estimate phase analysis in alloy such as M23C6, M6C carbides and Ti (N, C)?
- In Figure 11(a) to (d), the Cr enriched phase is Cr23C6, and Ti-, and N-rich particles are Ti (N, C). In (f) to (h), they are considered to be carbides but cannot be identified as M23C6 or M6C because of the nano size.
- The caption in Figure 11 was revised to make this clear: “Figure 11. As‑welded fusion zone using Haynes 230 filler material showing the different carbides. (a) and (e) HAADF images, (b)-(d) ChemiSTEM EDX maps of Cr, Ti and N indicating MC (Ti-rich) and M23C6 (Cr-rich) carbides, and (f)-(h) ChemiSTEM EDX maps showing nano‑sized W, Mn and Cr-rich carbides. Note that their type did not be identified because of the nano size.”
- Note that the words with the bold font were added.
Line 282, 284, 285, and 286: Correct and write Figure/cross-reference?
- The “Error! Reference source not found” has been corrected. It should be “Figure 14”
Line 315: What is the morphology and shape of M23C6?
- The following sentences were revised to add the morphology and shape of M23C6: “They appeared in the three main manners: 1) blocky M23C6 (arrowed in (a)) segregated in the grain boundary and had a size of hundreds of nano‑meters, Figure 17(a); 2) cube-shape M23C6 clustered adjacent to grain boundary and also precipitated along the slip planes or dislocations, and having a size of about 50 nm and containing Moiré fringes, as shown in the enlarged area in (b). The amount of these M23C6 gradually decreased towards the grain interiors, Figure 17(b); 3) cube-shape M23C6 cluster consisted of large blocky M23C6 (hundreds of nms in size) and smaller ones (about 50 nm in size and containing Moiré fringes), Figure 17(c)”.
- Note that the words with the bold font were added.
Line 317: Could the authors show where Moiré fringes located relating to grain boundary?
- The cube-shape M23C6, having a size of about 50 nm, clustered adjacent to grain boundary and they contain Moiré fringes. An enlarged area was added in Figure 17 (b) to show the Moiré fringes. Moiré fringes is not related to grain boundary.
Line 321: Could you show where theses particles placed?
- Revised the sentence and added “(arrowed in (d))” to make it clear “Figure 17(d) provides an enlarged TEM BF image showing small M23C6 about 50 nm in size (arrowed in (d)) containing Moiré fringes and Oroman dislocation loops with Ti (C, N) precipitates.”
- Note that the words with the bold font were added.
Line 327: Do you mean 'types' instead 'tyoes'?
- Sorry, “tyoes” is a typo. It has been corrected to “types”.
Round 2
Reviewer 1 Report
Comments and Suggestions for Authors
The reviewer buys the authors' argument, but think that they should accommodate the question/answers in the revised manuscript, in additional to those editorial changes.
Author Response
Thanks for the reviewer's feedback. We believe we have addressed all the comments and made changes accordingly in the manuscript.
Thanks again for the time spent on reviewing.